Transcriptomic characterization revealed that METTL7A inhibits melanoma progression via the p53 signaling pathway and immunomodulatory pathway

Zhang Duoli 1
Zou Tao 1
Liu Qingsong 2
Chen Jie 1
Xiao Mintao 1
Zheng Anfu 1
Zhang Zhuo 1
Du Fukuan 1 3 4
Dai Yalan 5
Xiang Shixin 6
Wu Xu 1 3 4
Li Mingxing 1 3 4
Chen Yu 1 3 4
Zhao Yueshui 1 3 4
Shen Jing 1 3 4
Chen Guiquan chenguiquan1973@126.com 7
Xiao Zhangang zhangangxiao@swmu.edu.cn 1 3 4
1 Department of Pharmacology, School of Pharmacy, Southwest Medical University, Laboratory of Molecular Pharmacology , Luzhou , China
2 Department of Pathology, The First People’s Hospital of Neijiang , Neijiang , China
3 Cell Therapy & Cell Drugs of Luzhou Key Laboratory , Luzhou , China
4 South Sichuan Institute of Translational Medicine , Luzhou , Sichuan , China
5 Department of Oncology, Affiliated Hospital of Southwest Medical University , Luzhou , Sichuan , China
6 Department of Pharmacy, University-Town Hospital of Chongqing Medical University , Chongqing , China
7 Chinese Medicine Hospital Affiliated to Southwest Medical University , Luzhou , Sichuan , China
Pickett Brett
Electronic publication date: 2023 Aug 2
Publication date: 2023
Volume: 11
Electronic Location ID: e15799
Received 2023 Mar 7; Accepted 2023 Jul 5
Copyright: ©2023 Zhang et al.
Copyright year: 2023
Copyright holder: Zhang et al.
License: This is an open access article distributed under the terms of the Creative Commons Attribution License, which permits unrestricted use, distribution, reproduction and adaptation in any medium and for any purpose provided that it is properly attributed. For attribution, the original author(s), title, publication source (PeerJ) and either DOI or URL of the article must be cited.
License URL: https://creativecommons.org/licenses/by/4.0/

Keywords: p53 signaling pathway, METTL7A, Melanoma, Immunomodulation

Funding: National Natural Science Foundation of China 81972643 82172962 Sichuan Science and Technology Project 2021YJ0201 Luxian People’s Government and Southwest Medical University Scientific and Technological Achievements Transfer and Transformation Strategic Cooperation Project 2019LXXNYKD-07 Science and Technology Program of Luzhou, China 21CGZHPT0001 This work was supported by the National Natural Science Foundation of China (No. 81972643, No. 82172962), the Sichuan Science and Technology Project (2021YJ0201), the Luxian People’s Government and Southwest Medical University Scientific and Technological Achievements Transfer and Transformation Strategic Cooperation Project (2019LXXNYKD-07) and the Science and Technology Program of Luzhou, China (No. 21CGZHPT0001). The funders had no role in study design, data collection and analysis, decision to publish, or preparation of the manuscript.

==============================
METTL7A is a protein-coding gene expected to be associated with methylation, and its expression disorder is associated with a range of diseases. However, few research have been carried out to explore the relationship between METTL7A and tumor malignant phenotype as well as the involvement potential mechanism. We conducted our research via a combination of silico analysis and molecular biology techniques to investigate the biological function of METTL7A in the progression of cancer. Gene expression and clinical information were extracted from the TCGA database to explore expression variation and prognostic value of METTL7A. In vitro, CCK8, transwell, wound healing and colony formation assays were conducted to explore the biological functions of METT7A in cancer cell. GSEA was performed to explore the signaling pathway involved in METTL7A and validated via western blotting. In conclusion, METTL7A was downregulated in most cancer tissues and its low expression was associated with shorter overall survival. In melanoma, METTL7A downregulation was associated with poorer clinical staging, lower levels of TIL infiltration, higher IC50 levels of chemotherapeutic agents, and poorer immunotherapy outcomes. QPCR results confirm that METTL7A is down-regulated in melanoma cells. Cell function assays showed that METTL7A knockdown promoted proliferation, invasion, migration and clone formation of melanoma cells. Mechanistic studies showed that METTL7A inhibits tumorigenicity through the p53 signaling pathway. Meanwhile, METTL7A is also a potential immune regulatory factor.

Introduction

Melanoma is a type of skin cancer with a high mortality rate, particularly prevalent in European, North American, and Oceanian countries (Zeng et al., 2022). The occurremce of melanoma is associated with prolonged UV exposure, which usually leads to the development of DNA damage (Pecorelli & Valacchi, 2022). Excessive tumor mutational burden (TMB) and high immunogenicity are among the most striking features of melanoma. In addition, melanoma is a enormously aggressive tumor which can effortlessly metastasize for the duration of the physique by means of lymphatic and vascular routes (Chen et al., 2022b). The five-year survival rate for metastatic melanoma is solely 4.6%, making early diagnosis particularly important (Xie et al., 2022). While immunotherapies such as PD-1 inhibitors have revolutionized the treatment of melanoma, the occurrence of primary and acquired drug resistance limits their initial and sustained benefits as therapeutic options. (Huang & Zappasodi, 2022). In summary, there is an urgent need to develop a new diagnostic and therapeutic biomarker.

Human methyltransferase-like proteins are part of a large family of proteins characterized by the presence of a binding domain for S-adenosylmethionine, a co-substrate for methylation reactions (Ignatova et al., 2019). Some members of this protein family have been estimated or proven to be DNA, RNA or protein methyltransferases. Previous studies have shown that members of the METTL family participate in a series of biological processes. For example, METTL3, METTL16, METTL2B, and METTL8 have been identified as RNA methyltransferases and play a critical role in tumor development. In particular, METTL3 and METTL14, which are “writers” in the m6A regulatory complex, have been shown to drive cancer development by forming a heterodimer that initiates m6A RNA methylation modifications (Chen et al., 2022a; Dou et al., 2023; Liu et al., 2022; Tooley, Catlin & Tooley, 2023). However, the biological function of a significant proportion of members is still unknown as of today. METTL7A, additionally acknowledged as AAMB, was once at first recognized as a lipid droplet-associated protein in Chinese hamster ovary K2 cells (Liu et al., 2004). Dysregulated METTL7A expression has been reported to be associated with a range of diseases. Overexpression of METTL7A enhanced osteogenic differentiation and viability of human bone marrow stem cells under glucose-free conditions as reported by Lee et al. (2021). Nain et al. (2021) determined downregulation of METTL7A expression in sufferers with COVID-19 patient. Wang et al. (2021a) confirmed that METTL7A promotes odontogenic differentiation of human dental pulp stem cells. In addition, abnormal expression of METTL7A has been detected in thyroid (Zhou et al., 2017), choriocarcinoma (Jun et al., 2020), gastric (Sexton et al., 2020), osteosarcoma (Jia et al., 2021), Lung adenocarcinoma (Guo, Ma & Zhou, 2019), breast (McKinnon & Mellor, 2017), and colorectal cancers (Lacalamita et al., 2021).

Numerous reviews have proven that loss of p53 signaling pathway feature would supply most cancers cells a survival advantage, permitting them to ignore oncogenic signals and the abrogation of DNA damage and proceed to proliferate abnormally (Hernandez Borrero & El-Deiry, 2021). This signaling pathway is involved in several biological effects to suppress cancer, including cell cycle, DNA replication and repair, cell proliferation, apoptosis, angiogenesis, and cellular stress response (Liebl & Hofmann, 2021). The transcription factor p53, the most essential central hub in this signaling, regulates these biological processes through transcriptional regulation of a series of downstream goal genes (El-Deiry, 1998; Tokino & Nakamura, 2000). p53 is altered (mutated and deleted) in half of the cancers and loses its capability to restrict cell proliferation, thereby advertising tumorigenesis or progression (Huang, 2021). p21 is a well-established inhibitor of cell cycle protein-dependent kinases (CDK) directly regulated by p53 transcription. Cyclin D is a regulator of CDKs, which promotes cell proliferation by driving the G1 phase of the cell cycle into the S phase (Wang et al., 2021b). Many studies have shown that p21 can inhibit cell proliferation by binding to Cyclin D and thereby inhibiting the kinase activity of the complex and blocking the G1 to S transition (Dai et al., 2013).

In this study, we explored the potential role of METTL7A in cancer progression through a combination of biological and molecular biology techniques. Our study aims to provide novel and effective biomarkers for cancer diagnosis, prognostic staging, and treatment. The workflow of this study is shown in Fig. 1.

Figure 1 Workflow of this study.

Materials and Methods

Data collection

Gene expression profiles, clinical follow-up (Table S1), copy number variation (CNV) and single nucleotide polymorphism (SNP) information were extracted from the The Cancer Genome Consortium (TCGA) database (https://portal.gdc.cancer.gov/) through the “TCGAbiolinks” package in R. To compare the difference in METTL7A expression between normal and tumor tissues, we extracted normal tissue sample information from the Genotype-Tissue Expression (GTEx) database (https://www.gtexportal.org/home/index.html). Cell line expression data were obtained from the CCLE database (https://sites.broadinstitute.org/ccle/) through “depmap” package. The external validation datasets GSE46517, GSE65904 and ICGC-SKCM-US were downloaded from the Gene Expression Omnibus (GEO) (https://www.ncbi.nlm.nih.gov/geo/) and Internationla Cancer Genome Consortium (ICGC) (https://dcc.icgc.org) (Table 1). Expression profiles were converted to the Transcripts per kilobase of exon model per Million mapped reads (TPM) and normalized by log2. Data mining of public databases was conducted as of April 2022.

Table 1 Melanoma external datasets summary.

Dataset ID	Sample number	GPL Platform ID	Sample type	
GSE56417	121	GPL96	104 Tumor Samples	
			17 Normal Samples	
GSE65904	214	GPL10558	With clinical information	
				
ICGC-SKCM-US	430	NULL	With clinical information	

Gene expression analysis

To investigate whether METTL7A is involved in the process of carcinogenesis, we compared the expression of METTL7A between tumor tissues and their adjacent normal tissues. The data was statistically analyzed and visualized using “ggplot2” in R.

Protein–protein interecation

A protein interaction network was conducted using STRING to identify the proteins with which METTL7A interacts and to uncover the roles that METTL7A plays in biological signaling, gene expression regulation, and other life processes. Only interactions with a minimum score of 0.4 were considered, and disconnected nodes in the network were excluded.

Prognostic analysis

The Kaplan–Meier prognostic analysis and univariate Cox regression model analysis were conducted to investigate in the relationship between METTL7A and overall survival through the “survival” package in R. HR (Hazard ratio) less than 1 was regarded as a favorable factor.

Genomic viariation analysis

Mutation annotation format with single nucleotide polymorphism (SNP) information were analyzed and visualized through “maftools” package. Copy number variation data was processed, analyzed, and visualized using the “ggplot2” package in the R project.

Functional enrichment analysis

The “clusterProfiler” and “GSVA” packages were utilized to investigate the specific mechanisms through which METTL7A participates in biological processes. The c2.cp.kegg.v7.5.1.entrez.gmt file was used to calculate differences in signaling pathways’ enrichment scores in various populations of the two cohorts. The gene set (c2.cp.kegg.v7.5.1.entrez.gmt) employed for the enrichment analysis was downloaded from MSigDB (https://www.gsea-msigdb.org/gsea/index.jsp). Additionally, we leveraged KOBAS (KEGG Orthology Based Annotation System, https://bio.tools/kobas), a comprehensive online enrichment analysis tool, to identify downstream signaling pathways and biological processes involved in METTL7A regulation from both GO and KEGG databases. Functional and pathway items with a significance level (p-value) less than 0.05 were deemed to indicate significant differences.

Drug sensitivity prediction

The oncoPredict package in R is based on cell line screening data to predict response to commonly used chemotherapy and targeted drugs developed by Maeser, Gruener & Huang (2021) to explore and discover new Biomarkers. A matrix of cell line expressions and a matrix of drug treatment information were extracted from the Genomics of Drug Sensitivity in Cancer (GDSC) (https://www.cancerrxgene.org/) and the Cancer Therapeutics Response Portal (CTRP) (http://portals.broadinstitute.org/ctrp/) databases respectively to train the model, and then cancer gene expression profiles were read to calculate the sample drug IC50.

Immuno-infiltration analysis

We investigated the relationship between METTL7A and the tumor microenvironment using various algorithms including EPIC, TIMER, quanTiseq, MCPcounter, and ESTIMATE. Based on a set of hot tumor signature genes (CXCR4, CD274, CXCL9, CXCL10, CXCL11, CD4, CXCR3, CD3E, CCL5, PDCD1, CD8A and CD8B), we divided the melanoma samples into hot and cold tumor samples via “ConsensusCluster” package in R project. We utilized the Tumor Immune Dysfunction and Exclusion (TIDE) algorithm (http://tide.dfci.harvard.edu/) to explore the potential of METTL7A as a predictor of immunotherapeutic response.

Cell culture

The PIG1 cells were purchased from Otwo Biotech (ShenZhen Inc. China), A2058, and SKMEL28 cells were purchased from the Otwo Biotech (ShenZhen) Inc. They were cultured in the Dulbecco’s modified Eagle’s medium (DMEM) (Gibco, Billings, MT, USA) and containing 10% Fetal Bovine Serum (FBS) (Gibco) in incubator with 37 °C, and consisted of 5% CO2 in the incubator.

Knockdown and overexpression of METTL7A

To knock down and overexpression the gene expression of METTL7A, siRNA (siMETTL7A-1-F: 5′-GCCCUCAUAUCUAUGGAUATT-3′, siMETTL7A-1-R: 5′-UAUCCAUAGAU AUGAGGGCTT-3′. siMETTL7A-2-F: 5′-CACUGUGAUAUACAACGAATT-3′, siMETTL7A-2-R: 5′-UUCGUUGUAUAUCACAGUGTT-3′), and overexpression plasmid (PLV-3FLAG-METTL7A) transfection were used. According to this study, the A2058 and SKMEL28 cells were treated with thin-strand interfering RNA (siRNA) and overexpression plasmid using jet-PRIME transfection reagent (Polyplus transfection, Illkirch-Graffenstaden, France). An overnight incubation at 37 °C was performed on the A2058 and SKMEL28 cells cultivated in a six-well plate. For each well, 200 µL jet buffer, 1 µL siRNA (100 µM) or 1 µg overexpression plasmid, and 4 µL jet-PRIME were mixed, after incubated at room temperature for 10 min then added to the cells. A 48-hour time period was used to collect transfected cells for further analysis. There were five groups after transfection in total: siRNA knockdown group: siMETTL7A-1, siMETTL7A-2, and their blank control(siNC); overexpression plasmid group: ov-METTL7A, and its empty plasmid control group.

RT-qPCR

In order to verify the transfection of efficiency and the expression of METTL7A in PIG1, A2058, and SKMEL28, RT-qPCR was used. TRIzol reagent (Thermo Fisher Scientific, Inc., Waltham, MA, USA) was used to extract total RNA from the 6-well plate, for the detection of METTL7A mRNA expression, reverse transcription-quantitative polymerase chain reaction (RT-qPCR) was used. In this study, qPCR was conducted on a CFX96 Touch Real-Time PCR system with a SYBR Green Real-Time PCR kit (Thermo Fisher Scientific, Inc.). The primers used for real-time PCR were METTL7A-F: 5′- GTGCAACCTGACCAGAGAGA -3′and METTL7A-R: 5′- GTGCTGCAGCTTCAGCTTAG -3′; GAPDH-F: 5′-CTGGGCTACACTGAGCACC-3′and GAPDH-R: 5′-AAGTGGTCGTTGAGGGCAATG-3′.In this study, qPCR was performed under the following conditions: 95 °C for 30s, followed by 40 cycles of 95 °C for 5 s, 55 °C for 30 s and 72 °C for 30 s. and the 2−ΔΔCq method was used to calculate changes in gene expression within a target gene.

Cell proliferation assay

Cell proliferation assay was used to test the influence of METTL7A in A2058 and SKMEL28 cells. When the cells were transfected 48 h later, 1,500 cells were plated in 96-well plates. To each well, 10 µL Cell Counting Kit8 (DojinDo, Kumamoto, Japan) was added after 24, 48, 72, and 96 h. Microplate readers (SpectraMax Plus 384; Molecular Devices, San Jose, CA, USA) were used to measure absorbance at 450 nm.

Cell invasion and migration assays

In this study, transwell filters (JETBIOFIL) were used to detect the cell invasion and migration of A2058 and SKMEL28. For cell migration assay, after A2058 and SKMEL28 cells were transfected for 48 h, cells were resuspended in 150 mL of serum-free medium in the upper chamber, and 600 mL of 10% FBS medium in the bottom chamber. After 48 h, cotton swabs were used to remove cells in the upper chamber. Cells that had invaded and migrated were placed on the outer side of the chamber, then fixed with 4% paraformaldehyde, stained with 0.1% crystal violet, and photographed by an inverted microscope (NIKON). In addition, we used transwell filters coated with Matrigel (Corning, Corning, NY, USA) on the supper surface of chambers for cell invasion assay. Three fields were randomly selected for counting.

Colony formation assay

To further prove the effectiveness of METTL7A on the proliferation capacity of A2058 and SKMEL28, the colony formation assay was used. Briefly, after transfected for 48 h, 500 cells of A2058 and SKMEL28 were plated in 6-well plates. 14 days later, 4% paraformaldehyde-fixed (BL539A; Biosharp, Hefei, China) and 0.05% crystal violet-stained (G1062; Solarbio, Beijing, China) cells were examined, repeated three times for counting.

Western blotting assay

In this study, the A2058 and SKMEL28 cells were lysed with RIPA lysis buffer (Beyotime, Jiangsu, China) at 4 °C for 30 min after the cells were transfected, which contain protease inhibitors (Roche, Indianapolis, IN, USA), and the BCA kit (Beyotime) was used to measure the protein concentration. Ten lane-volumes of proteins were separated using 10% SDS-PAGE and transferred to a polyvinylidene difluoride membrane (0.22 µm; Millipore, Billerica, MA, USA). The membranes blocked with 5% nonfat milk (LP0033; Oxoid, Basingstoke, Hampshire, UK) for 1 h at room temperature, and incubated the membrane with relevant antibodies overnight at 4 °C: against METTL7A (1:1000, GTX65969; Genetex, Irvine, CA, USA), THBS1 (1:1000, 18304-1-AP; Proteintech, Rosemont, IL, USA), p53 (1:1000, 250143; ZenBio, Durham, NC, USA), p21 (1:1000, 2947; CST, Danvers, MA, USA), Cyclin D1 (1:1000, CST, 55506) and GAPDH (1:1000, GTX100118; Genetex). All antibodies were diluted with 5% Bovine Serum Albumin (WXBC4157V; Vetec, Hohenhameln, Germany). After being washed, the membrane was incubated with secondary antibodies (goat anti-mouse HRP (1:3000, A0563; Beyotime) or goat anti-rabbit HRP (1:3000, A0516; Beyotime). Secondary antibodies are diluted with 5% nonfat milk. A chromogenic substrate (Bio-Rad, Hercules, CA, USA) such as ECL is used for visualization.

Statistical analysis

Experimental data are presented as mean ±standard error. Protein band quantification of western blotting was performed using ImageJ software. Analysis and visualization were conducted using Prism 7.0 (GraphPad, San Diego, CA, USA) and R 4.1.1 (R Core Team, 2021) project. Student’s t-test and Wilcoxon test were used to compare differences between two groups, while one-way ANOVA and Kruskal-Wallis tests were used to compare differences between more than two groups. Pearson test was used for correlation analysis.

Results

METTL7A is downregulated in most tumor tissues and its downregulation is related to shorter overall survival

METTL7A was down-regulated in most cancer tissues and has minimal expression in HNSC and SKCM (Fig. 2A). Among 28 tumor cell lines, we found that METTL7A expression was lowest in melanoma, renal cancer, liposarcoma and adrenal cancer cell lines (Fig. S2A). Kaplan–Meier analysis confirmed that high METTL7A expression was related to longer overall survival in most cancer types (Fig. 2B, Fig. S1). Meanwhile, univariate cox regression model analysis also revealed that METTL7A was a favorable prognostic factor in most types of human cancers including melanoma (HR < 1 and CI < 0) (Fig. 2C). METTL7A has protein interactions with cancer-related genes BCAS1 and GLIPR1L2, as well as post-transcriptional regulatory genes METTL8 and DDX55 (Fig. 2D). At the genomic level, the mutation rate of METTL7A was low (Fig. S2B) and copy number variation were mainly characterized by copy number amplification, although the rate of change was low (Fig. S2C). Although METTL7A expression was the lowest in HNSC, it was not associated with overall survival in HNSC (pvalue >0.05). Therefore, we focused subsequent analysis on melanoma, as METTL7A had the lowest expression level in both HNSC and melanoma.

Figure 2 In most human cancers, METTL7A is often downregulated in tumor tissues and its excessive expression is substantially related with extended OS.

(A) METTL7A is often downregulated in tumor tissues. (B) Prognostic analysis suggests that excessive expression favors extended OS. (C) Univariate Cox analysis indicated that METTL7A was a favorable prognostic element in most human most cancer types. (D) Proteins that interact with METTL7A.

Downregulated of METTL7A was related to poorer clinical staging in melanoma

Through external datasets and qPCR assays, we found that METTL7A was downregulated in melanoma cell or tissue and METTL7A has a high level of diagnostic ability for melanoma (Figs. 3A–3C). In more than one cohort, we additionally found that METTL7A upregulation was related to extended OS and progression free survival (PFS) (Fig. 3D). Breslow depth is an indicator of the extent of melanoma metastasis, and its value is proportional to the extent of metastasis. Our silico analysis confirmed that METTL7A was negatively correlated with breslow depth and was down-regulated in poorer clinical staging (Tumor topography and survival time) (Figs. 3E–3F). These results further confirm that METTL7A is a cancer suppressor gene in melanoma.

Figure 3 METTL7A lower expression population is related with poorer clinical staging.

(A) In the GSE46517 cohort, METTL7A expression was upregulated in normal tissues. (B) METTL7A has an excessive stage of diagnostic cost value for melanoma in more than onecohort. (C) qPCR assay confirmed that METTL7A was upregulated in GIP1 PIG1 of normal skin cells in contrast to melanoma cells. (D) High METTL7A expression in a couple of cohorts was once determined to be considerably related to extended OS, PFS. PFS: Time between the start of randomisation and the onset of (any aspect of) progression of the tumour tumor or death (from any cause). OS, Overall survival. (E) Expression level of METTL7A was negatively related with breslow depth. (F) METTL7A lower expression population is related to poorer clinical staging.

METTL7A inhibited the proliferation, invasion, migration, and colony formation of A2058 and SKMEL28 cells

To investigate the impact of METTL7A on the malignant cell phenotype, melanoma cell function experiments were conducted in vitro. Firstly, we constructed knockdown and overexpression METTL7A tumor cell models by transfection, and verified the transfection efficiency by qPCR and Western blot (Figs. 4A, 5C). After knockdown of METTL7A, we observed a significant promotion of the proliferation, invasion, migration, and colony formation of A2058 and SKMEL28 melanoma cells compared to the control group (Figs. 4B–4D). Conversely, an increase in METTL7A expression would reverse above trends.

Figure 4 METTL7A inhibited the proliferation, invasion, migration, and colony formation of A2058 and SKMEL28 cells.

Here are five groups: siRNA knockdown group: siMETTL7A-1, siMETTL7A-2, and their blank control (siNC); the overexpression plasmid group: ov-ME; the overexpression plasmid group: ov-METTL7A, and its empty plasmid control group. (A) After transfection, the transfection efficiency of METTL7A was verified by qPCR in A2058 and SKMEL28. (B) In A2058 and SKMEL28, knockdown of METTL7A promotes tumor cell proliferation. (C) In A2058 and SKMEL28, knockdown of METTL7A promotes tumor cell invasion and migration. (D) In A2058 and SKMEL28, knockdown of METTL7A promotes tumor cell clone formation.

Figure 5 METTL7A inhibits melanoma progression via p53/p21/Cyclin D and p53/TSP1 axes.

(A) GO functional analysis revealed that METTL7A is involved in many immune-related biology process. (B) KEGG signaling pathway analysis revealed that METTL7A is involved in a lot of immune-related pathways and p53 signaling pathway. (C) GSEA revealed that MTTL7A low expression population was significantly associated with inhibition of immune-related pathways as well as the p53 signaling pathway. (D) In TCGA cohort, METTL7A is positively related with TP53. (E) Western blotting includes a total of five groups: siRNA knockdown group: siMETTL7A-1, siMETTL7A-2, and their blank control (siNC); the overexpression plasmid group: ov-METTL7A, and its empty plasmid control group. Western blotting comfirmed that altered METTL7A expression would lead to altered expression of key protein nodes in the p53 signaling pathway.

METTL7A can activate immune-related pathways as well as the P53 signaling pathway

Go functional enrichment analysis revealed that METTL7A is involved in a number of immune-related biological processes (Fig. 5A), meanwhile KEGG pathway enrichment analysis revealed that METTL7A is involved in immune-related pathways as well as the p53 signaling pathway (Fig. 5B). To investigate the relationship between altered METTL7A expression and activation or inhibition of signaling pathways in greater detail, we conducted GSEA. Given that METTL7A was down-regulated in tumor patients, we used the METTL7A low expression population as the case group and the METTL7A high expression population as the control group. Pathway enrichment analysis revealed that METTL7A downregulation inhibits immune-related pathways as well as the P53 pathway (NES = −1.606, P-value = 0.034) (Fig. 5C) (Fig. S4). In the cohort of TCGA, we observed a similar trend in mRNA expression changes for both METTL7A and TP53 (Fig. 5D). The same trend in protein expression changes for both METTL7A and P53 was also detected through Western blotting. In the si-METTL7A group, we also observed alterations in other proteins involved in the p53 signaling pathway, including downregulation of p21 and TSP1, and upregulation of Cyclin D1. The contrary end result was detected in the METTL7A overexpression group. The results above suggest that METTL7A is associated with the activation of the p53 signaling pathway (Fig. 5E). Mechanistically, METTL7A inhibits melanoma cell proliferation through the p53/p21/Cyclin D1 axis. Additionally, downregulation of TSP1 was observed in the si-METTL7A group. It is a key balance in the angiogenic switch can bind to transmembrane receptors displayed by endothelial cells, thereby inducing inhibition of angiogenic signaling (Kazerounian, Yee & Lawler, 2008), has been shown in previous in vitro and in vivo studies to inhibit melanoma progression in a variety of ways (Trotter, Colwell & Tron, 2003).

METTL7A can be used to predict the responsiveness of immunotherapy as nicely as chemotherapy in melanoma

The pathway enrichment analysis above revealed that downregulation of METTL7A was linked to the inhibition of immune-related pathways, indicating that METTL7A might impact cancer progression through immune pathways. To further investigate the connection between METTL7A and the tumor microenvironment, we utilized several algorithms, including EPIC, TIMER, quanTiseq and MCPcounter. These algorithms consistently demonstrated that downregulation of METTL7A was significantly associated with reduced levels of TIL infiltration (Fig. 6A). On the other hand, the ESTIMATE algorithm revealed that METTL7A downregulation was significantly related to lower immune, stromal and estimate scores (Fig. 6B). The above findings suggest that METTL7A may have the potential to remodel the tumor microenvironment in a positive way. Previous clinical studies have demonstrated that TILs possess the ability to eliminate tumors, and that patients with higher levels of TILs in their tumor tissue tend to have better prognoses. This partly explains why upregulation of METTL7A is associated with extended overall survival (Archilla-Ortega et al., 2022). In addition, the increase in TIL was reported to have a significant improvement in the response of patients to immunotherapy (Moore et al., 2022; Qiao et al., 2022). Thus, we postulated that upregulation of METTL7A could enhance the efficacy of immunotherapy. The TIDE algorithm analysis indicated that individuals with upregulated METTL7A expression may have a greater likelihood of responding to immunotherapy (Fig. 6C) (p value <  0.0001). We employed the R package “oncoPredict” to evaluate the sensitivity of four drugs frequently utilized in melanoma treatment. The results indicated that the high METTL7A expressing population had lower IC50 values for these drugs, indicating a greater susceptibility to chemotherapy and molecular targeted therapy (Fig. 6D). Hot tumors are those infiltrated with many anti-tumor T lymphocytes and tend to be more responsive to immunotherapy (Ren et al., 2022). We extracted a set of hot tumor signature genes from a previous study (Dong et al., 2021) and distinguished melanoma samples into hot and cold tumor samples by using of hot tumor signature genes (CXCR4, CD274, CXCL9, CXCL10, CXCL11, CD4, CXCR3, CD3E, CCL5, PDCD1, CD8A and CD8B). Ten hot tumor characteristic genes exhibit significantly different expression patterns in cold and hot tumor samples (Fig. 6E). METTL7A is upregulated in hot tumors (Fig. 6F). Within a certain range, patients with high METTL7A expression and concurrent hot tumors exhibited the longest overall survival, whereas those with low METTL7A expression and concurrent cold tumors experienced the shortest overall survival (Fig. 6G). METTL7A expression also possesses moderately effective diagnostic value for distinguishing between hot and cold tumor samples (Fig. 6H). This result was also observed in the ICGC-US-SKCM cohort (Fig. S3).

Figure 6 In melanoma, METTL7A is a potential immunomodulatory factor.

(A) Higher levels of TIL infiltration in people with high METTL7A expression. (B) Higher levels of stromal, immune and estimation scores in populations with high METTL7A expression. (C) Lower IC50 values for commonly used chemotherapeutic agents in a population with high METTL7A expression. (D) Higher response rates to immunotherapy are predicted in populations with high METTL7A expression. (E) Melanoma patients were divided into hot tumor samples and cold tumor samples by a set of hot tumor signature genes. The heatmap displays the expression of ten hot tumor characteristic genes in cold and hot tumor samples. (F) METTL7A is highly expressed in tumor samples. (G) In METTL7A high expression and hot tumor combined group, patients have the longest overall survival while patients in METTL7A low expression and cold tumor combined group have the lowest overall survival. (H) METTL7A has moderately efficient diagnostic value for hot and cold tumor samples.

Discussion

The methyltransferase-like family is a diverse group of methyltransferases that have been shown to be extensively involved in the methylation of nucleotides, proteins, and small molecules (Tooley, Catlin & Tooley, 2023). METTL7A is a relatively understudied protein-coding gene within this family. However, numerous studies have reported dysregulation of METTL7A expression in both oncological and non-oncological diseases, indicating a possible role in disease development. Despite this, there is limited research on the biological function of METTL7A in cancer cells, its mechanism of action, and its relationship with the tumor microenvironment. Therefore, in our study, we explored the role of METTL7A in cancer progression and its relationship with tumor microenvironment by a combination of bioinformatics analysis and molecular biology techniques.

Analysis of TCGA bulk-seq data revealed that dysregulation of METTL7A expression was prevalent in various human cancer tissues, with downregulation of METTL7A commonly observed in cancer tissues. This suggests that METTL7A may play a role in cancer progression. Further analysis of clinical follow-up data indicated that low expression of METTL7A was associated with decreased overall survival and poorer clinical staging. These findings highlight the potential of METTL7A as both a prognostic marker and a potential therapeutic target. In vitro qPCR experiments revealed that METTL7A is downregulated in melanoma cells compared to normal human skin cells. METTL7A downregulation promotes proliferation, invasion, migration and clone formation in melanoma cells. METTL7A overexpression reverses this phenomenon. Mechanistically, pathway enrichment reveals that METTL7A low expression population inhibits p53 signaling as well as immune-related pathways. p53 signaling pathway is closely associated with phenotypes that promote cancer progression, such as prolonged DNA damage, disruption of DNA repair mechanisms, cellular senescence, cell cycle, apoptosis, angiogenesis and tumor metastasis (Jha et al., 2022). Western blotting revealed that knocking down METTL7A downregulated p53, p21 expression and upregulated Cyclin D1 expression. p21 and Cyclin D1 are two cell cycle regulation-related proteins. Cumulative reports indicate that p53 inhibits the cell cycle from S-phase to G-phase by inducing p21 and subsequently inhibiting Cyclin D activity, ultimately suppressing tumor cell proliferation (Chen, Bargonetti & Prives, 1995; Guo et al., 2014; O’Connor et al., 2021). Furthermore, changes in METTL7A expression also modulate the expression of TSP1. Studies have shown reported that tumor angiogenesis as well as tumor metastasis could be prevented through the p53/TSP1 pathway (Giuriato et al., 2006; Trapp et al., 2010). Several studies have reported decreased survival rates observed in TSP1-deficient mice with lymphoma, sarcoma, melanoma (Lawler et al., 2001). Another study demonstrated that TSP1 deficiency promoted precocious progression of lung lesions to diffuse adenocarcinomas (Baek et al., 2013). In melanoma, knockdown of TSP1 resulted in a 68% increase in peritumor angiogenesis and higher tumor growth rate, suggesting a direct tumor suppressive effect of TSP1 (Lindner et al., 2013). Based on the aforementioned findings, it can be inferred that METTL7A suppresses the malignant progression of melanoma via the p53/p21/Cyclin D1 and p53/TSP1 pathways.

On the other hand, the pathway enrichment analysis also indicated that low expression of METTL7A was linked to the inhibition of immune-related pathways, suggesting a possible association between METTL7A and the tumor microenvironment. The tumor microenvironment is known as the “seventh signature feature” of tumors, mainly composed of intrinsic immune cells, adaptive immune cells, cytokines, cell surface molecules, etc. It has long been considered to be closely related to tumor development, recurrence, metastasis and tumor immunotherapy (Junttila & de Sauvage, 2013; Smyth et al., 2016). Tumor immunotherapy is a highly anticipated approach that encompasses four types of immune checkpoint inhibitors, cellular periprocedural immunotherapy, tumor vaccines, and non-specific immunomodulators. It has been utilized to treat various types of cancer, such as melanoma, non-small cell lung cancer, and kidney cancer (Kraehenbuehl et al., 2022; Marabelle et al., 2017; Zhang et al., 2022). Immune cells play an important role in the process of tumor immunotherapy. It has been reported that tumors are also referred to as hot and cold tumors depending on the degree of immune cell infiltration around the cancer cells, and they will have very different outcomes in response to immunotherapy (Galon & Bruni, 2019). Hot tumors, also known as immune inflammatory tumors, are characterized by high levels of T cell infiltration, elevated interferon-gamma signaling pathways, increased PD-L1 expression, and a high tumor mutation burden. These tumors are often more responsive to immune checkpoint inhibitors (ICIs) and exhibit greater efficacy. In contrast, cold tumors, also known as immune desert or immune excluded tumors, have TILs located at the edge of invasion or do not have TILs at all. These tumors are often insensitive to ICI treatment (Duan et al., 2020; Galon & Bruni, 2019). Our bioinformatics analysis identified METTL7A low expression population associated with inhibition of B-cell receptor signaling pathway, T-cell receptor signaling pathway, antigen processing and presentation, and primary immunodeficiency pathways. Primary immunodeficiency is a heterogeneous group of immunodeficiency disease, typically accompanied by a decrease in acquired immune cell count and/or function. The most common clinical manifestations are recurrent infections autoimmune diseases, and malignant tumor (Ballow, Sanchez-Ramon & Walter, 2022; Mohtashami et al., 2022). On the other hand, the strength of B- and T-cell receptor signaling correlates with T- and B-cell responses, and signal deficiency will lead to restricted T- and B-cell activation (Shakiba et al., 2022). Our silico analysis also showed that METTL7A high expression population was associated with higher levels of TIL and infiltration of immune cells including B cells and NK cells, and these data emphasize the potential of METTL7A to reshape the tumor microenvironment, although further proof of causality is lacking. Moreover, the TIDE algorithm and analysis of hot and cold tumors demonstrated that high METTL7A expression is associated with increased responsiveness and effectiveness of immunotherapy. These findings underscore the crucial role of METTL7A in the context of immunotherapy.

This study employed a combination of computational analysis and experimental validation to explore the biological functions and mechanisms of action of METTL7A in cancer progression and its interaction with the tumor microenvironment. The results showed that METTL7A presents a robust diagnostic, prognostic, and therapeutic biomarker in melanoma. However, our study was limited to bioinformatic computational analysis and in vitro experiments, and the role of METTL7A in vivo requires further exploration. Moreover, the correlation between METTL7A and the tumor microenvironment is strong, yet the causality of this relationship, as well as the specific mechanisms involved in this process, also require further investigation.

Conclusion

Our study demonstrates that METTL7A expression is dysregulated in various types of tumors, and increased expression is correlated with prolonged overall survival - consistent with previous research findings (Liu, Chen & Shen, 2023; Pan et al., 2022; Tooley, Catlin & Tooley, 2023). Moreover, in melanoma, our findings indicate that upregulation of METTL7A can suppress proliferation, invasion, migration, and colony formation in vitro. Mechanistic studies suggest that METTL7A inhibits melanoma progression via the p53 pathway. Additionally, METTL7A has the potential to modulate immunity and may serve as a predictive and therapeutic target for the immunotherapy.

Supplemental Information

Supplemental Information 1 Kaplan–Meier prognostic analysis of METTL7A in pan-cancer

Click here for additional data file.

Supplemental Information 2 Additional analysis of TCGA database information

(A) The expression level of METTL7A in different tumor cell lines. (B) Mutations of METTL7A in different cancer types. (C) Copy number variation of METTL7A in different cancer types.

Click here for additional data file.

Supplemental Information 3 In ICGC-cohort, METTL7A is a potential immunomodulatory factor

(A–D) Melanoma patients were divided into hot tumor samples and cold tumor samples by a set of hot tumor signature genes. (E) METTL7A is highly expressed in tumor samples. (F) In METTL7A high expression and hot tumor combined group, patients have the longest overall survival while patients in METTL7A low expression and cold tumor combined group have the lowest overall survival. (G) METTL7A has moderately efficient diagnostic value for hot and cold tumor samples.

Click here for additional data file.

Supplemental Information 4 Individual plots of significant pathways in GSEA analysis results

Click here for additional data file.

Supplemental Information 5 Clinical Characteristics of the study subjects

Click here for additional data file.

Supplemental Information 6 R code

Click here for additional data file.

Supplemental Information 7 Uncropped western blots for Fig. 5E

Click here for additional data file.

Supplemental Information 8 Raw Data

Click here for additional data file.

Abbreviations

ACC Adrenocortical Carcinoma

BLCA Bladder Urothelial Carcinoma

BRCA Breast Invasive Carcinoma

CESC Cervical Squamous Cell Carcinoma and Endocervical Adenocarcinoma

CHOL Cholangiocarcinoma

COAD Colon Adenocarcinoma

DLBC Lymphoid Neoplasm Diffuse Large B-cell Lymphoma

ESCA Esophageal Carcinoma

GBM Glioblastoma Multiforme

HNSC Head and Neck Squamous Cell Carcinoma

KICH Kidney Chromophobe

KIRC Kidney Renal Clear Cell Carcinoma

KIRP Kidney Renal Papillary Cell Carcinoma

LAML Acute Myeloid Leukemia

LGG Brain Lower Grade Glioma

LIHC Liver Hepatocellular Carcinoma

LUAD Lung Adenocarcinoma

LUSC Lung Squamous Cell Carcinoma

MESO Mesothelioma

OV Ovarian Serous Cystadenocarcinoma

PAAD Pancreatic Adenocarcinoma

PCPG Pheochromocytoma and Paraganglioma

PRAD Prostate Adenocarcinoma

READ Rectum Adenocarcinoma

SARC Sarcoma

SKCM Skin Cutaneous Melanoma

STAD Stomach Adenocarcinoma

TGCT Testicular Germ Cell Tumors

THCA Thyroid Carcinoma

THYM Thymoma

UCEC Uterine Corpus Endometrial Carcinoma

UCS Uterine Carcinosarcoma

UVM Uveal Melanoma

TCGA The Cancer Genome Atlas

GEO Gene Expression Omnibus

ICGC International Cancer Genome Consortium

GTEx Genotype-Tissue Expression

GSEA Gene Set Enrichment Analysis

TME Tumor Microenvironment

CTL Cytotoxic T Lymphocyte

TIL Tumor Infiltrating Lymphocyte

Th cell Helper T Cell

ICI Immune Checkpoint Inhibitor

CTRP Cancer Therapeutics Response Portal

GDSC Genomics of Drug Sensitivity in Cancer

TPM Transcripts Per Kilobase of Exon Model Per Million Mapped Reads

SNP Single Nucleotide Polymorphism

CNV Copy Number Variation

TIDE Tumor Immune Dysfunction and Exclusion

IC50 The Half Maximal Inhibitory Concentration

TSP1 Thrombospondin 1

OS Overall Survival

PFS Progression Free Survival

DSS Disease Free Survival

RT-qPCR Real-time Quantitative Polymerase Chain Reaction

siRNA Small Interfering RNA

CDK Cyclin-dependent Kinase

COVID-19 Corona Virus Disease 2019

Additional Information and Declarations

Competing Interests

Author Contributions

Data Availability

The authors declare there are no competing interests.

Duoli Zhang conceived and designed the experiments, performed the experiments, analyzed the data, prepared figures and/or tables, authored or reviewed drafts of the article, and approved the final draft.

Tao Zou conceived and designed the experiments, performed the experiments, analyzed the data, prepared figures and/or tables, authored or reviewed drafts of the article, and approved the final draft.

Qingsong Liu conceived and designed the experiments, authored or reviewed drafts of the article, and approved the final draft.

Jie Chen conceived and designed the experiments, performed the experiments, analyzed the data, prepared figures and/or tables, and approved the final draft.

Mintao Xiao conceived and designed the experiments, performed the experiments, analyzed the data, prepared figures and/or tables, authored or reviewed drafts of the article, and approved the final draft.

Anfu Zheng conceived and designed the experiments, performed the experiments, analyzed the data, prepared figures and/or tables, authored or reviewed drafts of the article, and approved the final draft.

Zhuo Zhang conceived and designed the experiments, performed the experiments, analyzed the data, prepared figures and/or tables, authored or reviewed drafts of the article, and approved the final draft.

Fukuan Du conceived and designed the experiments, performed the experiments, analyzed the data, prepared figures and/or tables, authored or reviewed drafts of the article, and approved the final draft.

Yalan Dai conceived and designed the experiments, performed the experiments, analyzed the data, prepared figures and/or tables, and approved the final draft.

Shixin Xiang conceived and designed the experiments, analyzed the data, prepared figures and/or tables, and approved the final draft.

Xu Wu conceived and designed the experiments, analyzed the data, prepared figures and/or tables, and approved the final draft.

Mingxing Li conceived and designed the experiments, analyzed the data, authored or reviewed drafts of the article, and approved the final draft.

Yu Chen conceived and designed the experiments, analyzed the data, authored or reviewed drafts of the article, and approved the final draft.

Yueshui Zhao conceived and designed the experiments, prepared figures and/or tables, authored or reviewed drafts of the article, and approved the final draft.

Jing Shen conceived and designed the experiments, prepared figures and/or tables, and approved the final draft.

Guiquan Chen conceived and designed the experiments, prepared figures and/or tables, and approved the final draft.

Zhangang Xiao conceived and designed the experiments, prepared figures and/or tables, and approved the final draft.

The following information was supplied regarding data availability:

The uncropped western blots, R code and raw data are available in the Supplemental Files.

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
