# Peer review of "Transcriptomic characterization revealed that METTL7A inhibits melanoma progression via the p53 signaling pathway and immunomodulatory pathway"

_PeerJ, doi:10.7717/peerj.15799_

## Round 0.1 · original submission · Major Revisions

Thank you for your submission to PeerJ. Overall, it seems the reviewers recommend clarifying parts of the text, softening some of the conclusions, and improving the display items. I would recommend that you adequately address each of the reviewers' comments if you decide to resubmit your manuscript for a second round of reviews.

Reviewer 1 ·

Basic reporting

This study investigated the biological function of METTL7A in cancer progression through a combination of bioinformatics analysis and molecular biology techniques. This is a very interesting topic, and the structure and argumentation of the paper have strong relevance, but some details may need further improvement by the author. In particular, the following issues need to be addressed:
①Abstract: There is a lot of research on the relationship between METTL7A and cancer, so it may not be appropriate to say that “However, few research have explored the relationship between METTL7A and cancer”.
②Introduction: The introduction only introduces the background, lacking key parts such as research purpose, significance, expected results, etc. It is recommended that the author supplement this part.
③Results: The expression "Considering that METTL7A expression in melanoma is low and associated with longer overall survival, we focused our subsequent analysis on melanoma. Line 252-254" contains ambiguity. The author points out that METTL7A expression is correlated with a better prognosis, but does not explain the trend of expression changes, which may mislead readers.

Experimental design

①Materials and methods: The author used multiple websites, but did not provide the websites and data retrieval time. It is recommended that the author supplement the relevant information in the appropriate places. The author's methodological description is not detailed enough, such as not specifying which package of R language was used for "Gene expression analysis"; and not explaining the specific parameters used when obtaining data for "Protein-protein interaction". The author is advised to provide additional information.
②Prognostic analysis: When defining protective factors, the author does not seem to consider the confidence interval (CI). Usually, only when HR is less than 1 and CI does not include 1, it can be considered a favorable factor. This issue also appears in figures 2B and 2D. It is recommended that the author supplement the confidence interval in the corresponding positions.
③Functional enrichment analysis: GSEA is a good enrichment analysis method, but using multiple enrichment methods may be more persuasive. It is recommended that the author supplement GO and KEGG enrichment analysis. In addition, the description of the GSEA enrichment analysis process is too general, and it is recommended that the author further supplement details, such as threshold information.

Validity of the findings

①The grouping is not clear, and there are inconsistencies between the context.
In Figure 2, the author analyzed METTL7A from the perspective of pan-cancer and used abbreviations for many cancer types, but did not provide explanations for these abbreviations in the text. The focus of this study is melanoma, and the lack of annotations may cause readers to be unclear about which item in Figure 2 represents melanoma.
The author constructed knockout and overexpression models of the METTL7A gene, but did not specify the names of the groups for each model. This issue is particularly evident in Figures 4 and 5.
The author stated in the methodology section in line 158 that PIG1 cells were used, but in Figure 3C, it changed to GIP1 cells. Why is there a difference in the cells used?
In lines 357-364, the article mentions TSP1, but in the figures, it is referred to as THBS1. Although these are two different names for the same substance, it can be misleading for readers who are not familiar with the field. The article does not explain the relationship between the two names. The author is advised to use consistent nomenclature in the figures and main text and provide explanations for any differences.
②Picture problems
In Figure 3B, there is no annotation for the confidence interval and p-value in both the figure and the main text. In the main text, there is also no description of Figure 3B. In addition, the x-axis of ROC curves is usually "1-Specificity (FPR)", why did the author choose “Specificity” as the x-axis?
In Figure 3D, we noticed that the number of individuals in the high and low expression groups of METTL7A is not the same, and it does not seem to be grouped based on the median. What is the basis for this grouping? Or could you explain why it was grouped this way?
According to lines 261-263 of the original text, " Our silico analysis confirmed that METTL7A was negatively correlated with Breslow depth and was down-regulated in poorer clinical staging (Figure 3E, Figure 3F)." However, the "Lymph node" and "Pathologic Stage" in Figure 3F do not seem to support this statement. What is the purpose of including these two figures in this context?
③The information provided in Figure 5A is insufficient. The NES and p-values alone cannot reflect the results of the enrichment analysis well. This issue is evident in lines 383-386. The author is advised to draw GSEA analysis plots for each pathway, based on the pathways that are the focus of the article, and provide necessary textual explanations in the article.
④It is recommended that the author supplement a table of clinical characteristics of the study subjects.

Reviewer 2 ·

Basic reporting

The manuscript „Transcriptomic characterization revealed that METTL7A inhibits melanoma progression via the p53 signaling pathway and immunomodulatory pathway“ by Zhang et al examines the role of METTL7A in melanoma using in silico and in vitro analysis. The authors reported the correlation of reduced expression of METTL7A with the cancer progression, shorter patient survival and worst clinical outcome. Authors performed in vitro analysis using two melanoma cell lines to investigate the biological function of METTL7A and reported increased proliferation, migration and invasion of melanoma cells after METTL7A KD, while the biological effects where reversed upon METTL7A overexpression pointing to the tumor suppressive role of METTL7A in melanoma. In addition, based on their findings, authors suggest that the tumor suppressive function of METTL7A is regulated via p53 signaling pathway and that METTL7A also exhibits potential immunomodulatory function. The prognostic relevance of METTL7A expression has already been reported for some cancers (PMID: 36830565, PMID: 36620541) and since here presented manuscript investigates the tumor suppressive function of METTL7A in melanoma holds a great promise. However, several major remarks need to be addresses and clarified before considering the manuscript for publication.

One of the major remarks concerns English language that requires extensive editing to make the manuscript understandable; thus, many sentences throughout the text should be rephrased/revised to make them clearer to the reader. Some examples are, but not limited to:
Lines 42-43: “On the other hand, METTL7A is additionally a potential immunomodulatory”
Lines 64-65: “Previous studies have shown that members of the METTL family are worried in a number of biological processes”
Lines 67-70: “In particular, METTL3 and METTL16, which are writers within m6A regulators, have been proven to mediate most cancers development by way of mediating m6A RNA methylation modification that adjust a number most cancers signaling” – please provide additional references that will support this statement, such as PMID: 36094754
Lines 92-93: “p21 is a well-established inhibitor of cell cycle protein-dependent kinases (CDI) direct regulated by p53 transcription” – please replace “direct” with “directly” and “CDI” with “CDK”
Lines 352-353: “Western blotting experiments Knockdown of METTL7A was shown to downregulate p53 and p21 in the p53 pathway and upregulate the expression levels of Cyclin D1.”
Lines 379-383: “Patients with hot tumors are treated with immune checkpoint inhibitors (ICI) when tumor-specific cells are activated and act as tumor killers, while patients with cold tumors have very little recognition of tumor cells due to immune cells, and therefore Immunotherapy is very ineffective, and typical cold tumors include breast, ovarian, prostate and pancreatic cancers(Duan et al. 2020; Galon & Bruni 2019)”.
Lines 386-388: “Frustration of primary immunodeficiency signaling will lead to impaired differentiation and maturation of immune cells, resulting in defective immune function and increased risk of malignancy(Ballow et al. 2022; Mohtashami et al. 2022).”
Lines 407-408: “Our study shows that METTL7A expression is disordered in a range of tumors and is associated with extended overall survival” – please rephrase as it is not clear whether increased or reduced METTL7A expression is associated with extended OS

The authors should cite several references to support their findings (PMID: 36094754, PMID: 36620541, PMID: 36830565)

Based on the references that are cited by the authors, the METTL16 should be replaced with METTL14 (line 68)

GIP1 should be replaced with PIG1 (line 174), and please replace GIP with PIG throughout the text

In the lines 282-286 authors mention two groups based on METTL7A protein expression and describe/designate as si-METTL7A and METTL7A overexpression groups which is quite confusing if it refers to the patient groups. Could authors please explain/comment?

There are several remarks that concern Figure 4, such as:
Second and fourth graph in Figure 4B seem identical
The description of the graphs in Figure 4 should be more detailed, especially in the Figure 4A. Based on the graphs on the right for both cell lines (Figure 4A), it seems that the siRNA mediated transfection increases the expression of METTL7A. Could authors please comment these findings? On the other hand, if the graphs show the METTL7A expression after overexpression then the title of the Figure 4 should be adjusted.
There is quite a difference between two control samples (in both cell line) for migration and invasion which one would not expect (Figure 4C and 4D). Could authors please comment these findings? Could authors please define control samples? Does control sample refers to untransfected cells or transfected with non-targeting siRNA/empty plasmid?

The major remark also concerns Figure 6 where detailed descriptions of the graphs are missing, especially for Figures 6E-6H.

Please revise the sentences due to typos that refer to the description of the Figures 2C, 5B and 5C. In addition, there are typos throughout the text (lines 352, 362, 381 – please replace capital letter with small letter and vice versa).

Several remarks concern incorrect citations/citing the references, such as:
Lines 357-358: “Giuriato and Trapp et al.” should be replaced with “Studies have shown”
The sentence (lines 359-361: “Lawler et al. showed in vivo tumorigenesis experiments using B16F10 melanoma cells that TSP1-deficient mice in vivo tumors grew at approximately twice the rate of controls (Giuriato et al. 2006)”) should include appropriate reference and should be revised to make it more understandable.
The beginning of the sentence (lines 361-362: “Lawler et al. also showed that endogenous TSP1 acts as a tumor growth inhibitor in vivo (Lawler 2002)”) should be rephrased since the reference (Lawler 2002) refers to the review article.

Experimental design

Relevant information (cat. no., manufacturer, etc.) about the siRNA targeting METTL7A should be provided in the Materials and Methods section

Validity of the findings

Based on the comparative in silico analysis the authors state that the METTL7A activates immune-related pathways and predicts the responsiveness of melanoma to immunotherapy, however the authors should include patient samples and/or RNAseq analysis upon METTL7A KD/overexpression in melanoma cell lines and/or in vivo models to get the results that will support such a bold statement. In my opinion the authors overinterpret/overestimate their data and thus, I would recommend the authors to tone down their conclusions focusing more on their results.

Annotated reviews are not available for download in order to protect the identity of reviewers who chose to remain anonymous.

---

## Round 0.2 · accepted · Accept

Thank you for addressing the reviewers' comments. I look forward to moving this manuscript through the publication process.

Reviewer 1 ·

Basic reporting

no comment

Experimental design

no comment

Validity of the findings

no comment

Additional comments

no comment

Reviewer 2 ·

Basic reporting

I would like to thank the authors for their responses and for improving their manuscript. I believe that the manuscript in the revised form meets the standard of PeerJ and would highly recommend it for the publication.

Experimental design

I would like to thank the authors for their responses and for improving their manuscript. I believe that the manuscript in the revised form meets the standard of PeerJ and would highly recommend it for the publication.

Validity of the findings

I would like to thank the authors for their responses and for improving their manuscript. I believe that the manuscript in the revised form meets the standard of PeerJ and would highly recommend it for the publication.